# Key Challenges in Diamond Coating of Titanium Implants: Current Status and Future Prospects

**DOI:** 10.3390/biomedicines10123149

**Published:** 2022-12-06

**Authors:** Maria Letizia Terranova

**Affiliations:** 1Dipartimento di Scienze e Tecnologie Chimiche, Università di Roma Tor Vergata, Via della Ricerca Scientifica, 00133 Roma, Italy; terranov@uniroma2.it or terranova@roma2.infn.it; 2Centro di Ricerca Interdipartimentale di Medicina Rigenerativa (CIMER), Università di Roma Tor Vergata, Via della Ricerca Scientifica, 00133 Roma, Italy

**Keywords:** diamond coatings, titanium, orthopedic implants, osteoblast, antimicrobial layer, sur-21 face treatments

## Abstract

Over past years, the fabrication of Ti-based permanent implants for fracture fixation, joint replacement and bone or tooth substitution, has become a routine task. However, it has been found that some degradation phenomena occurring on the Ti surface limits the life or the efficiency of the artificial constructs. The task of avoiding such adverse effects, to prevent microbial colonization and to accelerate osteointegration, is being faced by a variety of approaches in order to adapt Ti surfaces to the needs of osseous tissues. Among the large set of biocompatible materials proposed as an interface between Ti and the hosting tissue, diamond has been proven to offer bioactive and mechanical properties able to match the specific requirements of osteoblasts. Advances in material science and implant engineering are now enabling us to produce micro- or nano-crystalline diamond coatings on a variety of differently shaped Ti constructs. The aim of this paper is to provide an overview of the research currently ongoing in the field of diamond-coated orthopedic Ti implants and to examine the evolution of the concepts that are accelerating the full transition of such technology from the laboratory to clinical applications.

## 1. Introduction

The rapid increase in the aged population is creating an ever increasing demand for the replacement of dysfunctional bone tissues with artificial constructs.

Over the past decades, exciting advances in material science have enabled substantial improvements in prosthesis, fracture fixation and replacement components. Beyond efficacy, safety and reliability, the increased comfort of patients is also required. In this context, the intersection of material science and biomedicine is called upon to solve the pressing challenges of designing and producing ceramic, polymeric and metallic osteobiological grafts able to fully substitute for conventional allografts, autografts and xenografts [1]. In particular, there is an increasing demand in the segment of metal-based high-fatigue strength applications [2].

The wide range of metallic devices designed to restore or replace damaged or diseased hard tissues include orthopedic implants (knee and arm joints, cups for hips, finger parts) as well as components for dental restoration (crowns, copings, bridges). Nowadays, the production of such tools used in bone and dental implantation surgeries far exceeds any other biomaterial applications. As an example, each year in the United States, over 800,000 knees and 350,000 hip replacements are made, with a total of 2,244,587 hip and knee arthroplasty procedures performed between 2012 and 2020. In 2021, a cumulative procedural volume growth of 18.3% occurred compared with the previous year [3] and projections indicate a further rise, with the global market of orthopedic metallic implants expected to reach USD 46.5 billion by 2024 [4].

As regards to metals, the stringent criteria required by load-bearing applications limit the options to the few with mechanical properties similar to natural bone, i.e., stainless steel, cobalt-chromium, and titanium. Titanium (Ti) and its alloys are certainly the synthetic materials more frequently used for permanent implants. Ti is a metal with no allergenic or immunogenic effects and it is characterized by a strength-to-weight ratio that enables a variety of load-bearing applications [5]. Moreover, among the metals that can be used for implants, Ti has an elastic modulus closer to that of outer cortical bone. This helps in reducing the mechanical stresses that may occur at the interface between the orthopedic components and host tissues, with negative effects on osteointegration [5].

To allow integration of the implant into the host tissue, the cell’s metabolism requires a local microenvironment able to assure cellular communication, the transport of oxygen and nutrients, and the removal of wastes produced. These complex regenerative processes at the bone-to-implant interface depend also upon physical and mechanical cues. The cells receive mechanical stimuli from the artificial environment and translate them into biochemical signals. Such signals modulate several aspects of cellular behavior, including attachment, proliferation, and differentiation. The achievement of successful bone remodeling is currently driving research towards the development of innovative implant materials with engineered surfaces that mimic the natural extracellular matrix (ECM) [6,7,8].

From a biological point of view, even if Ti is not intrinsically an osteoinductive and osteoconductive material, the research of the last two decades have highlighted that effective osseointegration can be induced on Ti surfaces via proper modulation of the chemistry, topography and surface energy [9,10,11,12,13,14,15,16,17,18,19,20]. 

However, decades of implantation surgery have provided evidence that the body-fluids induce in Ti several adverse processes, such as wear, corrosion, and eventually, tribocorrosion [21].

These processes are particularly strong when Ti components are inserted into rather aggressive environments, such as the oral one [22,23,24]. 

Therefore, surface engineering of the Ti-based implants and the improvement of the interface between the implants and the host tissue have become a key task [5,14].

A great variety of mechanical, physical and chemical treatments have been proposed to modify the Ti surfaces and to control the finishing of the prostheses. These strategies range from sand-blasting using abrasive particles to etching induced by acids, high voltages or plasmas [18,25,26,27,28,29]. A rationale use of finishing treatments demonstrated the feasibility of generating Ti surfaces with a topography similar to that of natural ECM, therefore, able to promote the initial adherence and subsequent growth of cells. These treatments have now passed the research phase and are commonly applied to commercial dental implants. 

Nevertheless, it has been found that osteopromotion is not the only biological process taking advantage of the micro and nano features present on modified Ti surfaces. In effect, the rough surfaces create an environment that is also favorable to bacterial adherence and biofilm formation [30,31,32]. In addition, it is now known that early microbial colonization depends on a number of other factors, such as the surface potential and wettability [33,34]. A scheme of the complex multi-step process that leads to biofilm formation on solid surfaces is shown in Figure 1. 

Over the last two decades, a commonly employed strategy to prevent infections of periprosthetic joints has been the coating of Ti biotic surfaces with antimicrobial layers [35]. To this purpose, a series of different approaches have been taken into consideration. These range from the fabrication of antibiotic-loaded layers [21,33] to the use of non-antibiotic organic antimicrobial agents [36] and also of inorganic antimicrobial agents [37,38]. This last class includes either non-metallic (F, N) or metallic elements (Ag, Au, Cu, Ca, V, Zn), inserted onto the Ti surfaces by a variety of chemical or physical methodologies [39]. To avoid prosthetic joint infections, heavy metal-based strategies have also been devised. Recent studies have suggested that Ni, Fe, Mn, Ga, Ce, Se, Cs, Y and Pd could also be promising candidates for an antifouling functionalization of Ti surfaces [21]. 

However, several disadvantages have been highlighted in the various approaches aimed at preventing the spread of bacteria and biofilm formation. Among the more severe drawbacks, there are the adverse effects of antibiotics on tissue integration, development of antibiotic resistance, and cell damage caused by the nonantibiotic organic antimicrobial agents [33,40]. Moreover, several researchers have pointed out that when the Ti surfaces are modified by biocidal metals, the corrosion resistance of Ti can be strongly compromised [41]. This effect, first highlighted by Wan et al. [42], is due to galvanic processes occurring between Ti and metals with different electrochemical potentials [43].

The review of literature reveals that, in general, the enhancement of bone mineralization and the control of bacteria colonization, the two main factors that assure a good integration of implants, have been addressed separately [44]. This is undoubtedly a sign of how hard the task could be to promote osseointegration while simultaneously contrasting microbial activity. Only in the last few years has the development of bioactive multifunctional coatings become a leading-edge task [45,46,47]. 

From this perspective, ongoing research concerning the design of long-lasting implantable constructs is now focusing not only on simultaneous osseointegration improvement and biofouling contrast, but also on complying with the mechanical properties required for prolonged and intense use [48].

## 2. Diamonds for Implant Technology 

The latest advances in the engineering of orthopedic Ti-coatings are being provided by the use of diamond, a material characterized by chemical stability, high hardness, corrosion resistance, antiwear properties, biocompatibility and cytocompatible behavior [49,50,51].

Due to their fascinating physicochemical properties, diamond-based materials have been progressively applied in several biomedical areas, from drug delivery to cellular imaging, and from labelling to assembling of multifunctional scaffolds for tissue engineering [52]. 

Regarding this latter application, from the beginning of the 2000s it has been evidenced that diamond surfaces enhance and, above all, selectively control the osteoblast-like MG 63 cell growth [53,54,55,56]. This finding has prompted researchers to focus on the integration of diamond-based coatings into orthopedic and dental Ti implants. 

Studies on the colonization of artificial diamond surfaces by a series of cells, such as fibroblasts, osteoblasts and mesenchymal stem cells had evidenced that sp^3^-coordinated carbon, either in the form of polycrystalline diamond films or micro/ nano-sized diamond grains, behaves simultaneously as a promoter of osseointegration and as an antibacterial agent [44,57,58,59,60,61]. In this context, a number of in vitro and in vivo tests contributed to modify the concept that the cellular response at the implant–bone interface is influenced solely by the diamond’s surface topography [48]. Even if some properly shaped diamond surfaces have shown to have biocidal activity [62], surface energy, wettability and conductivity also proved to be crucial in promoting antimicrobial effects and reducing implant failure [48]. It is to be noted that the exact mechanism by which the surfaces of the various artificial diamonds induce resistance against bacterial colonization is still under debate. A survey of the more recently proposed mechanisms can be found in [63].

Moreover, to favor the bioactivity of the cells and to promote subsequent tissue integration, the mechanical properties of a material must match those of the tissue to which it is applied [5]. This requirement is critical if one wants to avoid, or at least mitigate, mechanical stresses at the implant–tissue interface. 

Investigations performed using different cell lines and substrates with various degrees of stiffness have demonstrated the dependence of cell morphology, motility, proliferation and differentiation on the elastic modulus of the substrate and on its adjustments [64,65,66,67]. The value of the substrate’s stiffness regulates the amplitude of tensile forces exerted by the cells on their adhesion sites, and consequently, determines the cell response.

The good compatibility of diamonds with MG 63 cells can be rationalized by considering that the mechanical properties of artificial diamond layers, even if dependent on their structure [68,69], match, in any case, rather closely with those of bone tissues [5,70]. 

Taken together, all of the studies strongly highlighted the potentialities of diamond as a favorable material for bone tissue–implant interfaces and evidenced that the cellular response to diamond is influenced by a complex interplay of texture, microstructure, surface chemistry and surface potential [71]. 

As regards the bioactive behavior of the inserted foreign materials, one must consider the differences among the chemical-physical-biological environments surrounding the various implants [33,72]. Dental devices, knee/arm joints, or rods for spinal fixations are different constructs that experience, indeed, very different physiological conditions. With this view, the cell–substrate interactions are influenced not only by the chemical-physical properties of the coatings but also by the stiffness of the surrounding tissue, the chemistry of the fluids, and the presence of different bacterial communities. This means that, in order to improve the applicability of diamond-Ti implants in different parts of a human body, the methodologies adopted must offer the feasibility to tune the mechanical and physical properties of the material as well as the chemical features of the surfaces contacting the host tissues. 

## 3. Strategies for Diamond Coating

The new scenario opened by diamond has led us to explore all of the options available to realize diamond–titanium systems. The engineering of Ti-diamond components can be accomplished by a variety of methods, from the synthesis of crystalline coatings using CVD-based techniques to the embedding of diamond nanopowders onto the surfaces of Ti implants.

### 3.1. Coating by CVD-Grown Diamond 

A widely adopted technique for Ti coating by diamond is the Chemical Vapor Deposition (CVD), which produces a diamond phase using gaseous mixtures of carbon-rich reactants and hydrogen activated by hot-filaments or Micro-Wave (MW) plasmas [73]. The CVD process enables the manufacture of polycrystalline diamond layers with grain sizes ranging from a few nanometers to several micrometers, making it possible to produce polycrystalline (grain sizes > 100 nm) and also nanocrystalline (grain sizes in the 5–100 nm range) layers [74,75,76]. 

Modulation of the layers’ texture and morphology can be achieved by varying the composition and flow rate of the gaseous phase. In particular, the addition of an inert gas during the CVD process increases the diamond nuclei density and enhances secondary nucleation, promoting the growth of nanosized grains [77]. The morphological and structural features of the diamond phase can also be modulated by means of specific treatments of the Ti surfaces. Sur et al. [78] successfully integrated nanocrystalline diamond layers on Ti surfaces preliminary treated by chemical and mechanical processes. The nanostructured surfaces promoted a dense Ti-oxide interlayer and enabled the growth of uniform diamond coatings that anchored to the substrate well. The combined action of Ti treatments and nano-diamond coatings was found to greatly improve the corrosion-resistance of Ti-based dental implants [78]. 

The deposition of crystalline diamond layers can be accomplished also on non-planar substrates. Figure 2 shows Ti curved surfaces simulating a temporomandibular joint (TMJ) dental implant, coated by a nanostructured diamond layer [79]. Tests performed on TMJ simulants coated by 3μ thick CVD-grown diamond demonstrated a hardness of about 60 GPa [79].

A mandibular movement simulator has been employed to perform functionality tests on diamond-coated Ti components that imitate condyle–fossa pairs. As reported in [80], the coated Ti constructs, measured after an equivalent of two years of use, evidenced a strongly improved wearing performance. 

The CVD technique also enables nanocrystalline diamond to be coated onto fiber-shaped substrates, as SiOx micro-fibers [81]. Mats of diamond-coated micro-fibers were proven to increase the elastic modulus by 3.7 times with respect to bare micro-fibers. This stiffer material is a promising substrate for bone tissue engineering. 

As a whole, the use of the CVD technique proved to be very effective for diamond coatings in implantology. This process of diamond deposition is, indeed, both resource-efficient and cost-efficient and enables a series of differently shaped substrates to be coated [82].

### 3.2. Coating by Detonation Nanodiamond 

The nanodiamond produced by detonation processes, commonly referred to as “detonation nanodiamond” (DND), was first explored in the 1980s [83]. As reported in the key review by Mochalin et al. [84], starting in the late 1990s, the outstanding properties of DND led to wider interest in these nanoparticles, that rapidly found applications in various biomedical fields. This nanomaterial consists of polyhedral diamond grains with sizes in the 5–10 nm range and is able to couple the well-known properties of bulk diamond with further outstanding features induced by its nanosized dimensions [84]. The surface electrostatic potential of the facets drives the self-assembly of the nanograins and the formation of strong agglomerates. Figure 3 shows the morphology of a typical DND powder.

Moreover the detonation nanodiamond is characterized by its ability to create a tunable surface through a number of easy functionalization processes [84,85]. As a whole, the DND can be terminated by single atoms, small functional groups, and also by biological macromolecules, such as vascular endothelial growth factors (VEGF) or deoxyribonucleic acid (DNA) [63,86].

The coupling of DND with Ti is mainly achieved by inserting the powders into polymeric matrices and by using such composite materials for the coating of implants [86,87]. Thin layers of pure DND can be deposited on Ti by means of the dip coating technique [45,88]. Even if this last approach produces coatings with lower compactness and structural uniformity, dip coating has been demonstrated as a reference technique when applying DND layers on additively manufactured (3D) medical implants. 

The additive manufacturing processes, that make it possible to create custom-made products by means of computer assisted technologies, are overcoming the limitations of standard techniques and are now finding wide applications in the field of implantology [89]. 

A detailed description of the additive manufacturing processes used for fabrication of Ti bone implants, along with some outstanding case studies, is given by Popov et al. [90]. In particular, the 3D printing of laser melted Ti is emerging as an exciting method to create customized orthopedic constructs with various geometries and to provide an effective solution for complex surgical cases [91].

Direct deposition and integration of nanodiamonds [92] or of polycrystalline diamond coatings [61,93,94] onto the surfaces of 3D pre-formed architectures have been found to enhance the interaction of osteoblasts with the Ti surfaces and, moreover, to inhibit bacterial colonization of subcutaneous implants [95,96].

## 4. Post-Processing and Surface Modifications 

Beyond their intrinsic extraordinary properties, diamond coatings can be further morphologically and chemically modified in order to best address the specific demands of biological tissues.

There are several ways to induce structural changes in the diamond surfaces and to produce substrates with a topography mimicking the main features of the cell microenvironment. Among a variety of settled techniques, the most efficient are molten salts etching [97], solid-phase metal contact [98,99], and plasma processing [100].

Plasma treatments represent a powerful strategy for modulating the topography of diamond layers. Arrays of vertically aligned nanowhiskers can be obtained in plasma-reactors by the H-etching of self-assembled detonation nanodiamonds [101]. As shown in Figure 4, MW-RF plasmas can also induce a nanostructuring of CVD-produced flat diamond surfaces, with the building of nanocones, nanorods and nanowhiskers [102]. 

Conversely, also surface smoothing can be achieved by proper plasma processes. As described in [103], the treatment of nanocrystalline diamond films by MW oxygen-plasma produced rather smooth surfaces, characterized by the presence of rounded features.

Plasma processes are also able to modify the surface chemistry of the diamond layers. Indeed, it is possible to saturate the dangling bonds of diamond with H and O atoms, or with NH_2_-groups [103].

It is interesting to note that modifying the surface chemistry induces changes in the value of the surface free energy and, therefore, heavily influences the biological response of the material. A series of in vitro studies evidenced strong differences in the behavior of cells contacting micropatterned diamond surfaces with specific chemical terminations.

A strong correlation between the surface free energy of diamond and osteoblast responses was evidenced in 2012 by Yang et al. [104]. In this study, the best integration of osteoblasts was obtained on diamond surfaces treated with NH_3_-plasmas, for which a high free energy (62.9 mN m^−1^) was measured; and the worst occurred on H-treated surfaces with lower surface free energy (40.2 mN m^−1^). This study established a monotonic correlation between surface free energy and the wettability of diamond and validated the dependence of the osteogenic potentialities of diamond on its surface functionalization.

After that, several other researches have been focused on the task of modulating the wettability of the coatings, developing plasma treatments able to generate diamond surfaces with either hydrophobic or hydrophilic characters, and tailoring the level of such properties [105]. Figure 5 shows the wetting behavior of diamond films before and after hydrogen (H-termination) and oxygen (O-termination) plasma treatments.

More recently, Broz et al. [50], using real-time imaging, evidenced how the initial cell distribution, rate of cell adhesion, distance of cell migration and cell proliferation are influenced by the different terminations of diamond surfaces exposed to O- or H-plasma. The findings of this research confirmed that the oxidized surfaces promote cell proliferation better than the hydrogenated ones [50]. The same conclusion was achieved by Stigler et al. [106], who employed immunohistochemistry to investigate cell proliferation on several differently terminated nanocrystalline diamond layers, grown on differently treated Ti substrates. The results of the measurements performed using surfaces with different roughness and different wettability (Figure 6) evidenced a relationship between cellular responses and the hydrophilicity of the oxidized surfaces.

Proteome studies of human osteoblast MG63 cells confirmed the potential of O- and NH_2_-termined hydrophilic nanocrystalline diamonds for osseointegration and bone formation [103,107]. The surface chemistry and wettability modified by oxygen and ammonia plasmas have been found to induce the upregulation of vimentin, cadherin and fibronectin, the proteins involved in the interactions between cells and the extracellular matrix [103]. 

In addition, Fong et al. [63], in their review of bone–implant interfaces, related the commonly noted enhancement of cell-to-surface adhesion and of cell-to-cell interactions to the increased hydrophilicity of processed polycrystalline diamond films.

Hydrophilicity is presently considered a key issue in implant engineering because it is related not only to high biocompatibility and to antibacterial effects, but also to a lower friction coefficient. This last factor, associated with wear behavior of the material, is important, especially for joint arthroplasty.

## 5. Addressing the Topic of Conductivity 

Beyond the topographical features and the surface chemistry, conductivity and surface charge also influence the cell–substrate interactions, mediating the initial plasma protein binding and impacting the whole growth process [5,108].

Surface conduction has long been recognized as a fundamental requirement for the adhesion and growth of cardiac and nervous cells [61]. However, in vitro studies have demonstrated that the growth of osteoblast-like cells is also greatly influenced by the conductive behavior of the materials used as scaffold. Grausova et al. [109] observed an increase in the numbers of human osteoblast-like MG 63 cells cultured on Boron-doped conductive nanocrystalline films with respect to those cultured on undoped insulating diamond substrates. The authors of the paper related the differences in the cell response to the differences in the distribution and organization of talin, a protein of focal adhesion plaques associated with cell adhesion receptors (Figure 7).

Therefore, a fundamental task in both regenerative medicine and implantation surgery is the modification of the diamond’s insulating characteristics. In the field of implantology, charged surfaces offer the additional benefit of strongly reducing bacterial adhesion [110]. 

Electrical conducting n-type diamond layers are obtained by default when the CVD-growth is carried out using a gaseous phase rich in Nitrogen (N), but a proper modulation of conductivity is commonly achieved by inserting Boron (B) inside the diamond lattice. The B-doping, which induces a p-type semiconductivity, is addressed by injecting B-containing gaseous compounds during the deposition process. Several studies have confirmed the capabilities of B- and N-doped diamonds to facilitate cellular adhesion and to contrast biofilm formation [111,112,113,114]. 

The possibility of inducing further functionalities, such as electrical conductivity, by adding a variety of foreign species to the diamond phase is an attractive side to CVD methodology [115]. To overcome possible drawbacks related to the presence of B, diamond layers designed for bio-related applications can also be made electrically conductive by the insertion of biocompatible metals. An example is Ti-doping, obtained by introducing into the CVD reactor metalorganic Ti-compounds in powder form, driven by inert gas fluxes [102]. This approach allows conductivity to be tuned and, at the same time, the texture of the diamond grains to be modulated [116]. The bioactivity of the Ti-doped diamond layers has been validated by in vitro testing of osteoblast-like cell growth [117]. The study evidenced that conductive Ti-doped diamonds improve the processes of cellular adhesion and proliferation with respect to the undoped insulating ones. The features of MG 63 cells grown for 4 days on undoped and Ti-doped diamond can be observed in Figure 8.

These findings are in line with those obtained by Fox et al. [118], who proposed a different approach to fabricating mixed Ti-diamond systems. Here, the diamond phase has not been applied as a coating, but conversely mixed with laser-fused Ti to produce a hybrid material used to fabricate 3D printed implants. The authors performed a detailed study on the dependence of the water contact angles, surface roughness and water droplets shapes on the diamond/Ti ratios of the composite materials. The insertion of diamond into the Ti matrix was found to produce surfaces with a 30% decrease in the water contact angle. These highly hydrophilic surfaces strongly increased mammalian cell adhesion and proliferation compared with the pure Ti surfaces.

## 6. Summary and Perspectives 

No other field of material science faces such a diverse set of challenges as the one of biomedical materials. A look at the huge number of recent papers dealing with orthopedic and dental implantology evidences a rapid evolution of this research area, with a trend following the advances in regenerative medicine and material engineering.

In a living organism, materials specifically designed to match the requirements of the biological environment can also be affected by the physicochemical reactions of tissues against a foreign body and can undergo severe degradation. This also applies to Ti, the leading material used in orthopedic implantology, which still had to face such difficulties from the beginning of its use in clinical practice. Among the Ti surface modifications proposed to prevent, or at least mitigate, adverse reactions, the most interesting could be the coating of Ti by the biocompatible and chemically inert diamond.

Diamond coating offers inherent protection against the common degradation phenomena induced in Ti by body fluids. However, it has been discovered that the chemical protection of Ti provided by the diamond coating does not automatically guarantee favorable outcomes. In effect, the diamond-coated orthopedic and dental replacements produced first resulted in poor osseointegration and infection occurrence.

The disappointing performance of the first diamond-coated Ti implants in promoting bone regeneration were rationalized when the progression of research highlighted that stiffness, micro or nano-roughness, and surface texture of the diamond could no longer be considered the only parameters triggering the attachment and proliferation of osteoblasts. 

Conversely, a deeper understanding of intercellular biochemical communications evidenced how a lot of additional features, including surface texture, topography and microstructure, surface chemistry, surface free energy and electroconductivity, contribute to the role of diamond as viable osteoblast–Ti interface. These indications pointed towards the need for precise engineering of the diamond surfaces to enhance osteointegration and reduce bacterial growth. However, looking through the literature, one can notice that, in general, the issues of osseointegration and that of biofouling mitigation have been addressed separately. Only in the last few years have proposed technological solutions been able to face these problems in a combined way. 

As briefly summarized in this review, a variety of chemical and physical approaches are being designed, nowadays, to adapt the features of the diamond coatings to use in Ti permanent implants. 

The control of the morphology, topography and surface texture of the diamond coatings is still an important task, which can be accomplished by varying the composition of the reactant vapor phase during the CVD processes or by performing specific pre-treatments of the Ti surfaces. In addition, the use of detonation nanodiamonds as the starting material enables the morphological features of the coatings to be controlled, either when the nanodiamond powders are inserted into the polymeric matrices or when a dip-coating methodology is used. 

Moreover, the controlled modification of diamond surfaces can be obtained via post-process protocols. Among the variety of chemical reactions developed to produce the surfaces and replicating, at best, the features of the extracellular matrix, plasma processing is certainly the leading technique for the micro- and nano-shaping of diamond layers. A further advantage of this technique is that proper plasma treatments make it possible to manipulate, beyond the topography, also the chemistry and therefore, the free surface energy of the coatings. This is a key point, because it is the value of this last parameter that manages the wettability and determines whether the applied coating will promote the attachment of cells to the implant. 

Nowadays, the generation of a surface terminated by specific atoms (mainly H and O) or groups (NH_2_-) is a deeply pursued task, because the effects of the hydrophilicity and hydrophobicity of materials on osteoblast adhesion and integration have become increasingly evident. Rather, it is now confirmed that oxidized diamond surfaces with high hydrophilicity are strongly requested in implantology, not only for improving cellular response, but also for counteracting biofouling and for reducing friction between the contacting surfaces.

Furthermore, electrical conductivity and the surface charge turned out to be relevant properties for the integration of implants in bone tissues. Conductive diamond layers have been proven to stimulate the interplay between cell-signaling factors, to enhance cells spreading and to accelerate osteointegration. Several techniques, *in primis* the CVD one, have been developed for diamond doping and for the production of layers with tunable conductivity. In addition to the methods that use B and N gaseous reactants to produce n- and p-type diamond layers, other modified CVD-based methodologies have the potential to modify the insulating character of diamond. In this last case, organometallic compounds are used to incorporate various foreign species, as in the biocompatible Ti. It is to be noted that, regardless of the inserted element, all of the conductive diamond layers assure the maintenance of cell viability and have revealed their ability to reduce the adhesion of bacterial colonies. Great potential for future applications has also been shown by alternative ways of obtaining implant conductivity, such as the 3D manufacturing of Ti fused phases mixed with diamond powders. 

Overall, from a systematic analysis of the current literature on implantology, there emerges the tendency to combine a variety of technologies to produce multifunctional diamond-on-Ti systems able to simultaneously promote osseointegration and inhibit microbial activity. 

The challenge is to manipulate the structure, to modulate the roughness, to trigger different surface chemistries, and to induce electrical conductivity while maintaining the diamond’s fundamental properties that favor the bioactivity of osteoblasts. Additional challenges are imposed by the different environments in which the cells live and by the different mechanical conditions of load-bearing (bones) and non load-bearing (dental) implants.

The combination of so many specificities requires the use of manufacturing approaches able to offer great flexibility in producing, engineering, and processing diamond coatings. It is evident that such a task has no single solution, and that any attempt to meet the challenging issues of osteogenesis, bacterial adhesion, and cell viability at the diamond-coated surfaces of implantable devices needs to put into play all of the techniques and protocols settled up until now and to envisage other hitherto unexplored approaches.

## Figures and Tables

**Figure 1 biomedicines-10-03149-f001:**
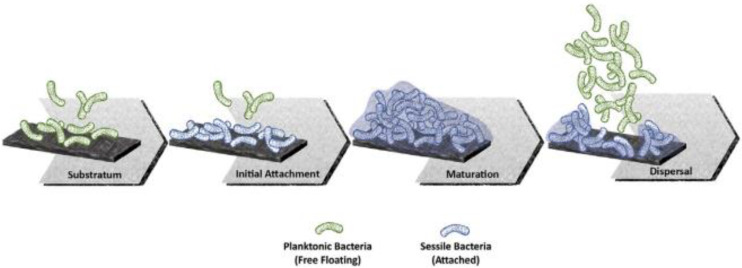
Schematic representation of the main stages of biofilm formation at the implant interface (From Ref. [32]).

**Figure 2 biomedicines-10-03149-f002:**
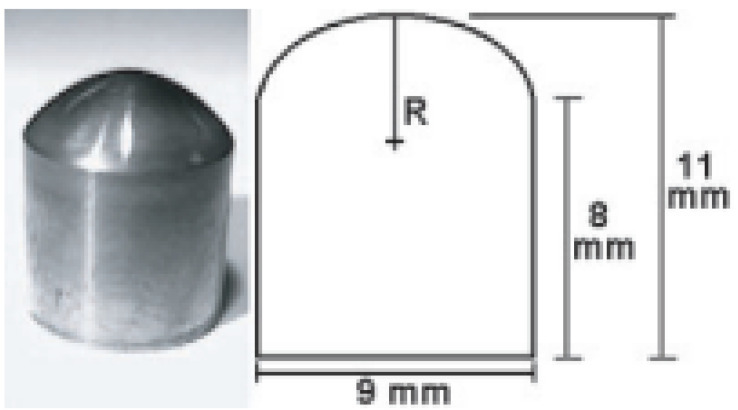
**Left**: Photo of a TMJ simulant with a 3µ thick nanostructured diamond coating. **Right**: dimensions of the TMJ simulant. The radius of curvature (R) is 6 mm (From Ref. [79]).

**Figure 3 biomedicines-10-03149-f003:**
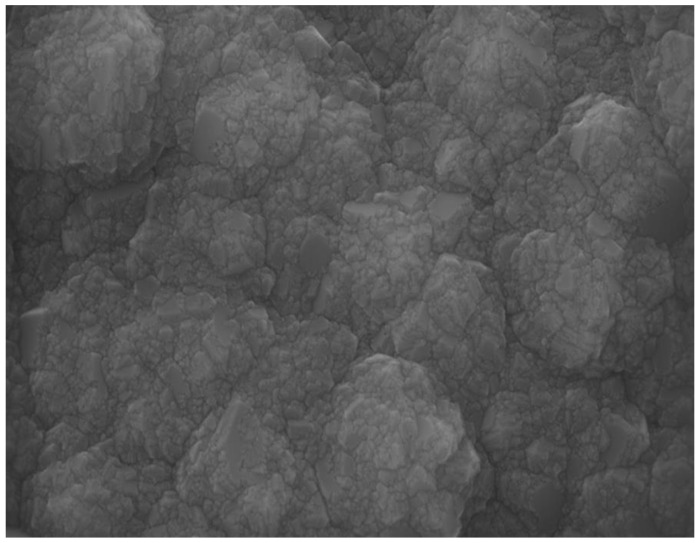
SEM image of a detonation nanodiamond powder, formed by aggregates of nanosized (<10 nm) polyhedral grains.

**Figure 4 biomedicines-10-03149-f004:**
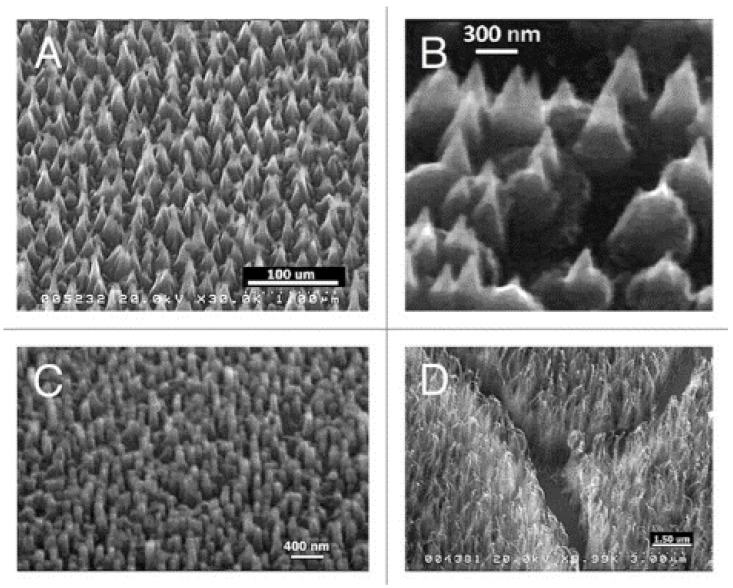
SEM images of diamond nanocones (**A**,**B**), nanopillars (**C**) and nanowhiskers (**D**), produced by the sculpturing of plane diamond films by means of a MW-RF CVD plasma reactor (From Ref. [102]).

**Figure 5 biomedicines-10-03149-f005:**
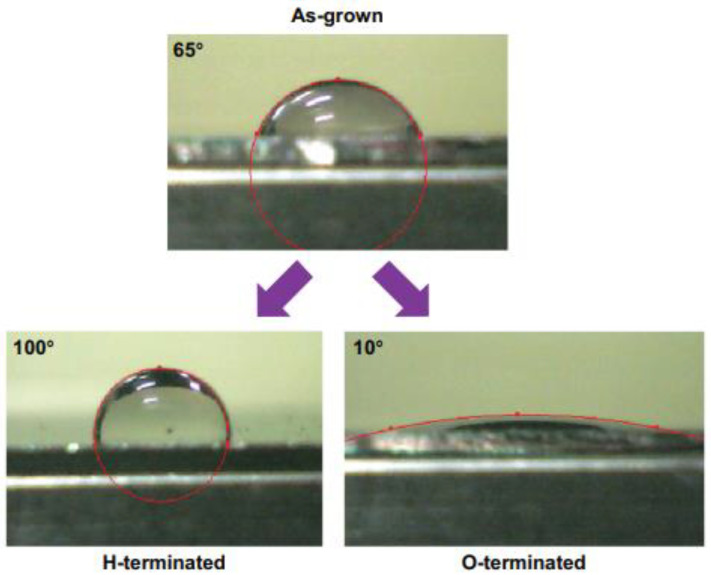
Wetting properties of diamond films as grown and treated in hydrogen plasma (H-terminated) or oxygen plasma (O-terminated) (From Ref. [105]).

**Figure 6 biomedicines-10-03149-f006:**
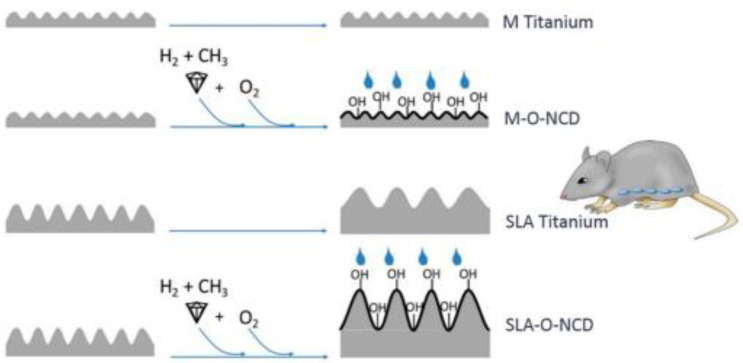
A design showing four different surfaces with different levels of roughness and different hydrophilicity achieved through additional nanosized diamond coatings; M Titanium: machined Titanium; M-O-NCD: O-terminated nanocrystalline diamond on machined Titanium; SLA Titanium: sand blasted and acid-etched Titanium; SLA-O-NCD: O-terminated nanocrystalline diamond on sand blasted and acid-etched Titanium (From Ref. [106]).

**Figure 7 biomedicines-10-03149-f007:**
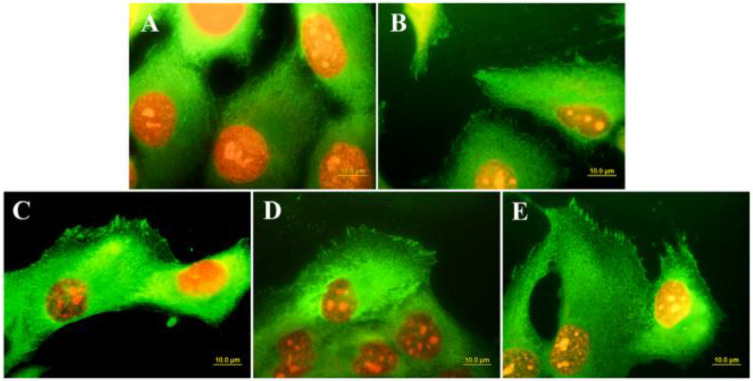
Immunofluorescence staining of talin in MG 63 cells on day 3 after seeding on microscopic glass coverslips (**A**), undoped nanocrystalline diamond (**B**), nanocrystalline diamond doped with Boron at various concentrations: 133 ppm (**C**); 1000 ppm (**D**); 6700 ppm (**E**). (From Ref. [109]).

**Figure 8 biomedicines-10-03149-f008:**
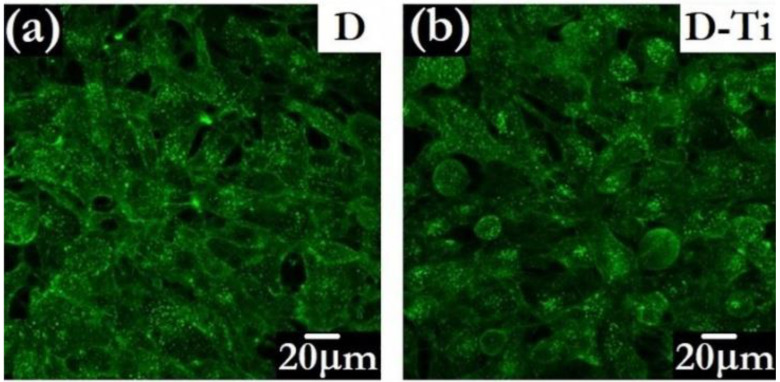
Confocal microscopy images of MG 63 cells grown for 4 days on undoped diamond D (**a**) and on Ti-doped D-Ti (**b**) samples (Adapted from Ref. [117]).

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
