# Peer review of "Key Challenges in Diamond Coating of Titanium Implants: Current Status and Future Prospects"

_biomedicines, 2022, doi:10.3390/biomedicines10123149_

Round 1
Reviewer 1 Report
This manuscript is well written and very clear. The topic is very interesting and innovative. This review discusses the diamond used as a biocompatible material used as an interface between titanium and tissue.
The diamond has adequate bioactive and mechanical properties. In fact, in this review the different experimental approaches for the pre-production of different diamond surfaces are described very well.
The introduction is very well written and detailed.
The different experimental approaches are well described. The figures are clear and well explained.
The final part of the review integrates and comments very well with the part written previously.
The bibliography is up-to-date and sufficiently numerous.
Reviewer 2 Report
The manuscript CANNOT be accepted for publication in its current form. It must be rewritten in a correct language level to be first understandable for the reviewer. Everything must be reorganized to be considered as a review paper. It is very difficult to evaluate it in its current form.
Reviewer 3 Report
The author formulated a review article on diamond-Ti systems that is interesting to read and well constructed. However, please see and address the following comments/questions:
1. Formatting inconsistency and/or issue: line 64; lines 82-83; line 110; line 165; line 250; lines 190-194 why highlighted?
2. Missing references: lines 96-98; lines 207-209; lines 266-268.
3. The article claimed to review research focusing on both osseointegration and biofouling mitigation but hardly mentioned any actual data on the latter matter. Please include some data/graphs on this topic.
4. The majority of this article focuses on the introduction and background given (sections 1, 2, and 3) rather than addressing the current challenges and how recent studies were conducted around such matters. Many references/researches in the articles are not recent enough as well. Please consider adding more recent data (2017 and newer).
5. The article was composed in a way of more spoken than written language. Please consider having the article proofread and edited by a professional to correct the grammar issues.
Round 2
Reviewer 2 Report
The revised manuscript needs several improvements to be accepted for publication. The mechanical stresses in the interface between implants and host tissue have a big effect in the osteointegration process. So, some references and comments must be added at least to the introduction. All abbreviations should be clarified from their first appearance (ex. MW ... etc.) or to add a table. Many English errors are found in the text. So, the manuscript must be revised before its acceptance for publication.
Additional comments:
Line 189, 'Sur et al. successfully ...'. Please add the reference to be 'Sur et al. [78] successfully ...'
In Line 196, 'In Fig.2 is displayed ...' It should be rewritten.
In Line 204, ' pairs were been carried out using' It should be rewritten.
Line 207, ' ... can be deposited by CVD also on ...' It should be rewritten.
Line 208, 'Diamond-coated SiOx micro-fiber mats, designed for potential bone tissue engineering, have proved to be a stiffer substrate , with an elastic modulus increased of about 3.7 times with respect to the bare microfibers'. It should be rewritten.
Line 216, 'Over the last years, the nanodiamond ...' What do the authors mean? 2, 3 years or more??? Reference is needed.
Line 233, 'A different approach is the dip coating technique , that enables to deposit on Ti components a thin layer of pure DND [45,88]' . It should be rewritten.
Line 238, More details and description about the different used additive manufacturing processes are needed.
Line 268, 'As described in [103] , nanocrystalline diamond films subjected to a MW O-plasma showed rather smooth surfaces , characterized by the presence of rounded features.' It should be rewritten.
Line 273, 'It is interesting to note that changes in the chemical termination not only modify the surface free energy, but influence also the biological response of the material.' It should be rewritten.
Line 319, 'Also Fong et al. in their review' Please add the reference to be 'Also Fong et al. [63] in their review'
Line 414, ‘Technological solutions able to tackle in a combined way these problems , major causes of implant failure and revision surgery, were proposed only in the last few years.’ It should be rewritten.
Reviewer 3 Report
Thank you, author(s) for addressing my previous comments/questions. No further comments on this article.
Round 3
Reviewer 2 Report
The authors answered the reviewer's questions. The manuscript can be accepted for publication after editing English language style.
